# Comparative Study of UV Radiation Resistance and Reactivation Characteristics of *E. coli* ATCC 8739 and Native Strains: Implications for Water Disinfection

Paola Duque-Sarango [1,2,*], Leonardo Romero-Martínez [3], Verónica Pinos-Vélez [2,4], Esteban Sánchez-Cordero [5,6] and Esteban Samaniego [2,6]

1   Grupo de Investigación en Recursos Hídricos (GIRH-UPS), Universidad Politécnica Salesiana, Campus El Vecino, Calle Vieja 12-30 y Elia Liut, Cuenca 010203, Ecuador
2   Departamento de Recursos Hídricos y Ciencias Ambientales, Eco-Campus Balzay, Av. Víctor Manuel Albornoz, Universidad de Cuenca, Cuenca 010203, Ecuador; veronica.pinos@ucuenca.edu.ec (V.P.-V.); esteban.samaniego@ucuenca.edu.ec (E.S.)
3   Departamento de Tecnologías del Medio Ambiente, Facultad de Ciencias del Mar y Ambientales, INMAR—Instituto Universitario de Investigación Marina, CEIMAR—Campus de Excelencia Internacional del Mar, Universidad de Cádiz, Campus de Puerto Real, 11510 Puerto Real, Spain; leonardo.romero@uca.es
4   Departamento de Biociencias, Eco-Campus Balzay, Universidad de Cuenca, Cuenca 010202, Ecuador
5   Departamento de Ingeniería Civil, Universidad de Cuenca, Cuenca 010203, Ecuador; esteban.sanchezc@ucuenca.edu.ec
6   Facultad de Ingeniería, Universidad de Cuenca, Av. 12 de abril s/n, Cuenca 010203, Ecuador
*   Correspondence: pduque@ups.edu.ec; Tel.: +593-0958923036

**Abstract:** In certain countries where fresh water is in short supply, the effluents from wastewater treatment plants are being recycled for other uses. For quality assurance, tertiary disinfection treatments are required. This study aims to evaluate the inactivating efficacy with an ultraviolet (UV) system on fecal bacteria from effluents of urban wastewater treatment facilities and the post-treatment influence of the environmental illumination. The effect from different UV doses was determined for native and standardized lyophilized strains of *Escherichia coli* right after the irradiation as well as after 24 h of incubation under light or dark conditions. To achieve 3 log-reductions of the initial bacterial concentration, a UV dose of approximately 12 mJ cm$^{-2}$ is needed for *E. coli* ATCC 8739 and native *E. coli*. However, there is a risk of the reactivation of 0.19% and 1.54% of the inactivated organisms, respectively, if the treated organisms are stored in an illuminated environment. This suggests that the post-treatment circumstances affect the treatment success; storing the treated water under an illuminated environment may pose a risk even if an effective inactivation was achieved during the irradiation.

**Keywords:** wastewater reuse; ultraviolet disinfection; flow-through UV reactor; photoreactivation; *Escherichia coli*

## 1. Introduction

Population growth, urban development, growing industries, and increased food production are straining freshwater resources. Urgent action is required to address the problem of water scarcity in today's globe. This threatens the short- and long-term health and safety of the global population. According to the World Health Organization, improved water resource management can reduce the global disease burden by 10% [1]. Therefore, treating drinking water adequately to meet a rapidly growing population's enormous needs is prudent. The United Nations endorsed a new agenda for sustainable development in 2015. Among its new Development Goals was the achievement of universal and equitable access to safe and affordable potable water for all by 2030 (SDG 6) [2]. It is estimated that at least 2 billion people live in countries with water scarcity, which will likely get

worse in some regions due to climate change and population growth. At least 2 billion individuals worldwide consume feces-contaminated drinking water. The most significant threat to the safety of potable water is microbiological contamination resulting from feces [1]. Microbiologically contaminated water can transmit diseases such as diarrhea, cholera, dysentery, typhoid fever, and polio and is estimated to cause 485,000 diarrhea-related fatalities annually [1].

An alternative to water demand is the reuse of treated wastewater effluent [3]. More and more countries use treated wastewater for irrigation, representing 7% of the irrigated land in developing countries. However, if wastewater treatment is carried out inappropriately, it presents health risks [1]. Different guidelines to regulate the reuse of wastewater have been published by national and international organizations such as the United States Environmental Protection Agency (USEPA) [4] and the World Health Organization (WHO) [5], which establish limits for the concentration of fecal coliforms.

Common bacteria associated with fecal microbiological water contamination include potentially pathogenic organisms such as *Escherichia coli*, intestinal enterococci, and *Clostridium*. Though present in the normal intestinal flora of humans and animals, some strains of these bacteria can cause infections when ingested and transported to other body regions. *E. coli* is a species of the genus *Escherichia*, which is predominantly composed of motile Gram-negative bacilli [6]. Some *E. coli* have acquired the ability to infect even the most robust human hosts with gastrointestinal, urinary, or central nervous system diseases [7]. Traditionally, *E. coli* has been used to monitor the presence of fecal contamination [5] and, consequently, the probable presence of pathogenic microorganisms [8]. International standards by the WHO and the European Union [5,9] designate *Escherichia coli*, *Enterococcus* spp. and thermo-resistant coliform bacteria as fecal indicators. Therefore, "non-detection" in 100 mL of sample is required to identify a water source as potable in all circumstances.

In most cases, disinfection is the last stage in drinking water treatment, and its purpose is to eliminate or inactivate pathogenic waterborne microorganisms. Disinfection can be achieved by physical and chemical methods that significantly reduce the total number of viable microorganisms in the water [10]. Typically, UV disinfection of water is applied by circulating the target water through tubes lined with UV lamps emitting light with a wavelength of approximately 254 nm [11]. The direct photochemical deterioration of nucleic acids [12] explains the germicidal effect of ultraviolet light. UV treatment has a considerable advantage over other technologies because, unlike many chemical disinfectants, it does not alter the taste or odor of water and poses no risk of overdose or by-product formation [13]. However, some microorganisms are known to be able to remediate UV-induced damage, which is the major challenge for this technology. Therefore, it is important to quantify the reactivation of microorganisms after treatment [14]. In photoreactivation, UV-induced DNA lesions are repaired by the photolyase, which requires near-ultraviolet light energy (310–480 nm) to be activated, whereas dark repair is independent of light [15]. These processes have received considerable attention because they can influence the efficacy of UV disinfection within hours of treatment, thereby jeopardizing the water's long-term safety.

The ATCC 8739 strain of *Escherichia coli* has been widely used as an indicator to validate the effectiveness of UV disinfection systems in various scientific studies [6]. This is because this strain is particularly resistant to UV radiation and is a good indicator of the efficacy of UV disinfection systems. In addition, the ATCC 8739 strain is also helpful because the processes of reactivation after exposure to UV radiation are more demanding compared to other bacterial strains, which means that if a UV disinfection system can successfully eliminate this strain, it is likely to be effective against different more sensitive strains of *E. coli* and other pathogenic bacteria [7].

It should be considered that bacteria in natural environments may develop a specific resistance to the UV disinfection processes [16]. Research in this field is mainly focused on the formal study of the dose processes necessary to inactivate laboratory strains of bacteria using pure cultures of microorganisms suspended in purified or deionized water; however, it would be interesting to study the response to UV treatment on isolated natural bacteria

and to compare inactivation efficiencies and quantify the reactivation processes, which could be significantly higher than for laboratory strains [15,17–20].

As a result, it would be helpful to establish the UV doses required for effectively inactivating native strains of fecal contamination as an indicator of treatment efficacy, analyzing the characteristics of higher or lower resistance to irradiation and lower or higher resistance to UV radiation compared to laboratory strains. This research discusses the survival behavior of natural *E. coli* versus a laboratory strain and analyzes the subsequent effects of reactivation in light and dark.

## 2. Materials and Methods

### 2.1. Organisms, Culturing Medium, and Pretreatment Incubation

Two types of microorganisms were used in this investigation: (1) a lyophilized strain of *E. coli* (ATCC 8739) obtained from the Spanish Type Culture Collection (CECT) and (2) *E. coli* bacteria isolated from the effluent of the Ucubamba wastewater treatment plant in the city of Cuenca, Ecuador.

The bacteria were stored as 50:50 glycerol–water suspension. For experiments, the organisms were reactivated by incubating them in 50 mL of Tryptic Soy Broth (TSB) culture medium (Sigma-Aldrich) for 24 h at 37 °C. Then, 1 mL of the culture was added to 50 mL of fresh TSB medium. After 24 h, the subculture was distributed in Eppendorf vials and centrifuged at 3000 rpm for 10 min; the supernatant was discarded and the pellet was washed with a 10% peptone solution, stirred, and added to 50 mL of phosphate-buffered distilled water at pH 7.20 to obtain the exponentially growing bacterial inoculum [21]. The inoculum was added to 20 L of pH 7.20 buffered distilled water in a drum to achieve concentrations of between $10^5$ and $10^6$ CFU mL$^{-1}$ (Figure 1).

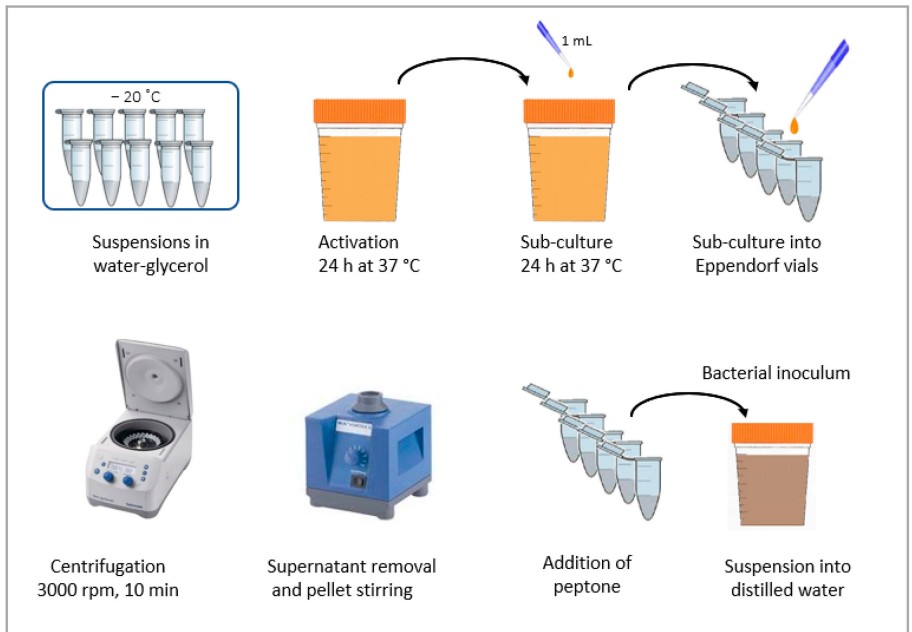

**Figure 1.** Reactivation of the glycerol–water preserved strain and preparation of bacterial inoculum for experimentation.

The membrane filtration method [22] was used, followed by an incubation period on chromogenic selective agar, Merck Coliform Agar Acc. Chromocult® in Petri dishes, and sterile 0.45 μm membrane filters (Pall Corporation, New York, NY, USA), to determine the bacterial concentration after treatment, expressed in CFUs (colony forming units). Each sample was diluted in decimal steps and seeded three times. The samples were incubated at 37 °C for 24 h. The generated CFUs were counted, and suitable dilutions were considered

to match plates with 20–150 CFU. During the microbiological analysis process, sterile conditions were ensured by using blank sample plates (Figure 2).

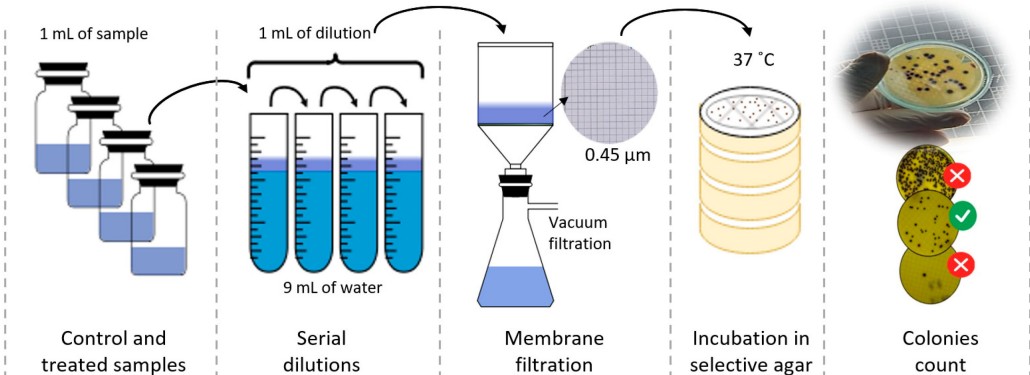

**Figure 2.** Post-treatment bacterial sample analysis and membrane filtration method. The number of colonies growing is adequate ($\sqrt{}$) or not adequate ($\times$).

### 2.2. UV Device Description and Dose Calculation

The UV treatment was applied by circulating the single-pass flow through the ring-type continuous flow reactor built at a laboratory scale. The equipment was composed of a 20 L capacity plastic tank with a centrifugal pump, manual valves to manipulate the flow rate, and a low-pressure mercury UV lamp (Phillips 1GPM—in/out 1/4″ monochromatic emission at 254 nm), covered by a quartz sleeve and inserted in an aluminum housing (Figure 3).

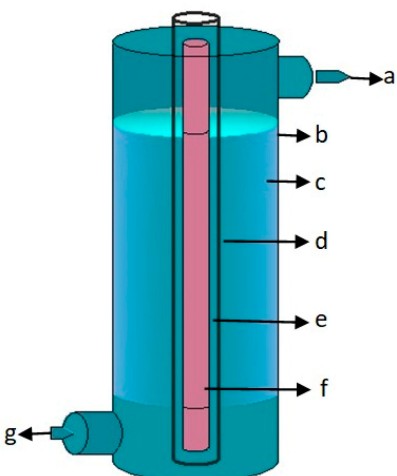

**Figure 3.** Elements of the low-pressure mercury UV lamp in the experiment. (a) Outlet, (b) Reactor case, (c) Water, (d) Quartz sleeve, (e) Air, (f) UV lamp, (g) Inlet.

According to USEPA guidelines [8], the UV dose applied (Equation (1); acronyms and values in Table 1) was calculated as the product of the mean intensity ($I_m$) and the theoretical retention time (*TRT*). The *TRT* was calculated as the quotient between the volume of water exposed to UV light ($V_R$) and the flow rate ($Q$). The $I_m$ was determined using the formulation of Equation (2) and a geometrical radiation scattering model for ring reactors previously supported by biodosimetry research [23,24]. At the start of each test, water transmittance ($T_W$) was determined using a Genesys 20 spectrophotometer set to 254 nm. The UV power at 254 nm, representing one-third of the total lamp power, was used to determine the average intensity ($P_{254}$) [25].

$$D = TRT \cdot I_m \qquad (1)$$

$$D = \frac{V_R}{Q} \cdot \frac{P_{254}\, T_Q^e}{2\, L\, \pi^2 \left(r_R^2 - r_Q^2\right)} \iint_{r_Q}^{r_R} \frac{T_W^{r-r_Q}}{r} \cdot dx\, dy \tag{2}$$

**Table 1.** UV lamp characteristics of the experimental system.

| Part | Parameter and Abbreviation | | Value |
|---|---|---|---|
| UV Lamp | Total power | $P$ | 6 W |
| | Power at 254 nm | $P_{254}$ | 2 W |
| | Length | $L$ | 18.9 cm |
| Quartz sleeve | Outer radius | $r_Q$ | 1.2 cm |
| | Thickness | $e$ | 1.6 mm |
| | Transmittance | $T_Q$ | 0.94 |
| Reactor case | Inner radius | $r_R$ | 2.55 cm |

### 2.3. Experimental Procedure

Before each test, the aseptic conditions of the working area and the experimental system were ensured. The materials and elements were cleaned and disinfected with hypochlorite and then rinsed with sterile water. The quartz sleeve was cleaned, and the UV lamp was switched on 5 min before the experiment to achieve stable radiation emission [26].

In the 20 L plastic tank, the stock solution containing the bacterial strains was placed and allowed to acclimatize for 30 min. The pump was then turned on to allow water to pass through the lamp at different flow rates. The exposure time was determined by measuring the flow rate with a 1 L graduated cylinder and a stopwatch. Once the flow rate was stabilized, a volume greater than the equipment's capacity was squandered before sample collection, ensuring that the sample was taken directly from the tank and contained the calculated dose. To prevent contamination of the reactor's downstream section, the UV doses were administered in descending order. Once the flow rate stabilized, a volume comparable to the total volume of the system was lost. Three samples were collected in sterile 250 mL vials per-flow rate. Immediately after UV irradiation (day 0), the contents of one of the three containers were subjected to the procedure to determine the bacterial concentration. The remaining two flasks were incubated for 24 h in a shaking incubator (model FS-70B) with a fluorescent light at 36 Einstein m$^{-2}$ s$^{-1}$ and 20 °C. One flask was covered with aluminum foil (1-day dark) (black container in the scheme), while the other one was left uncovered (1-day light) (blue container in the scheme). After incubation, the contents of both containers were subjected to the above-described procedure for determining bacterial concentration (see Figure 4).

### 2.4. Analysis of the Results

The level of inactivation achieved by UV treatment was calculated as log ($N/N_0$), where $N_0$ and $N$ are the number of colonies (CFU mL$^{-1}$) before and after UV irradiation, respectively. The inactivation curves were obtained by representing log ($N/N_0$) with respect to the applied dose, and this was fitted to inactivation models using the Geeraerd and Van Impe inactivation model-fitting tool (GInaFiT) [27]. As an estimate of treatment efficacy, the values of $D_3$ were determined as the UV dose required to reduce the initial concentration into three logarithmic units.

The photoreactivation percentage (PRP), which indicated the percentage of photoreactivated bacteria after 24 h among the bacteria affected by UV irradiation, was calculated according to Equation (3) [28]:

$$\text{PRP (\%)} = \frac{N_p - N}{N_0 - N} \times 100\% \tag{3}$$

where: $N_p$ = bacterial concentration of the photoreactivated sample (CFU mL$^{-1}$); $N$ = bacterial concentration immediately after UV disinfection (CFU mL$^{-1}$); $N_0$ = bacterial concentration before UV disinfection (CFU mL$^{-1}$).

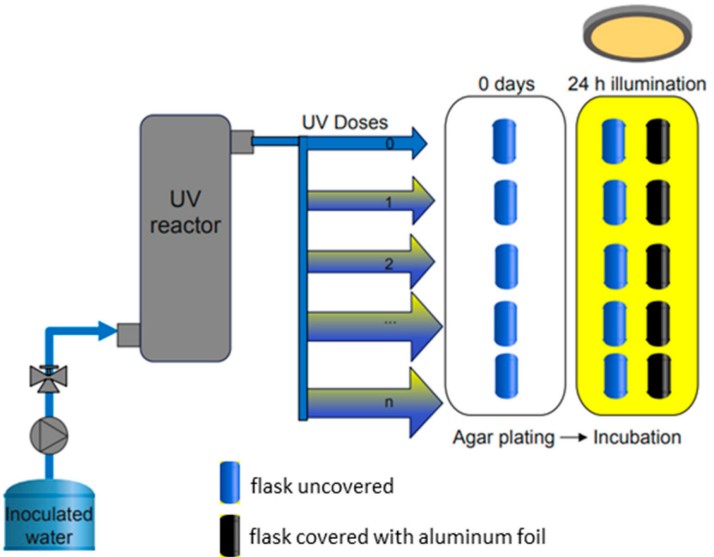

**Figure 4.** Scheme of the laboratory plant and experimental procedure.

## 3. Results

### 3.1. Inactivation Kinetics

The inactivation curves represent the values of Log (*S*) with respect to the applied UV dose (Figure 5). Data follow first-order kinetics at a low UV dose range, whereas they become asymptotic beyond a determined dose value.

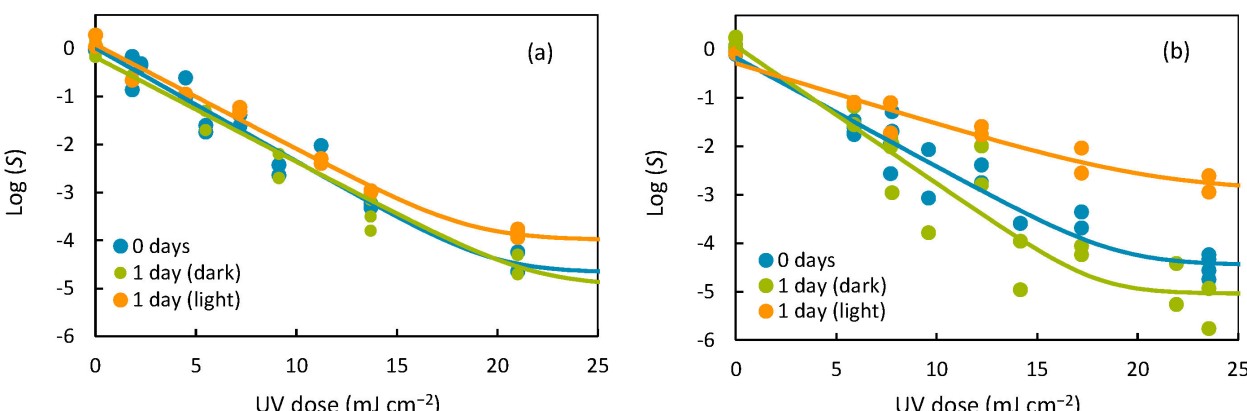

**Figure 5.** Inactivation curves of (**a**) *E. coli* (ATCC 8739) and (**b**) *E. coli* (native) for different post-treatment conditions. Symbols represent the experimental data and lines represent the inactivation model fitted (blue line: 0 d, green line: 1 d—dark, orange line: 1 d—light).

Experimental Log (*S*) data could be fitted according to a log-linear model with tail (Equation (4)), where *S* represents survival; $S_0$ represents survival in the absence of irradiation; $S_{res}$ represents residual survival; *k* represents inactivation rate, and *D* represents UV dose [29]. The R$^2$ values showed a strong correlation between Log (*S*) and UV doses, with values above 0.9 in most of the cases (Table 2).

$$S = (S_0 - S_{res})\, e^{-k \cdot D} + S_{res} \tag{4}$$

**Table 2.** Parameter values of the kinetic model for the two *E. coli* strains under different post-treatment conditions. "NR" indicates that the $D_3$ was not reached within the experimental range of UV doses.

| Organism | Treatment | $k \pm$ SE (cm$^2$ mJ$^{-1}$) | Log ($S_0$) $\pm$ SE | Log ($S_{res}$) $\pm$ SE | $R^2$ | RMSE | $D_3$ (mJ cm$^{-2}$) $\pm$ SE |
|---|---|---|---|---|---|---|---|
| *E. coli* (ATCC 8739) | 0 days | $0.54 \pm 0.03$ | $0.00 \pm 0.10$ | $-4.67 \pm 0.16$ | 0.972 | 0.302 | $12.8 \pm 1.3$ |
| | 1 day (dark) | $0.50 \pm 0.08$ | $-0.18 \pm 0.10$ | $-4.94 \pm 0.36$ | 0.937 | 0.511 | $12.1 \pm 1.8$ |
| | 1 day (light) | $0.50 \pm 0.02$ | $0.08 \pm 0.09$ | $-3.99 \pm 0.11$ | 0.990 | 0.166 | $14.4 \pm 1.2$ |
| *E. coli* (native) | 0 days | $0.52 \pm 0.04$ | $-0.17 \pm 0.16$ | $-4.45 \pm 0.21$ | 0.935 | 0.381 | $12.6 \pm 2.0$ |
| | 1 day (dark) | $0.65 \pm 0.08$ | $0.06 \pm 0.29$ | $-5.04 \pm 0.35$ | 0.882 | 0.673 | $10.8 \pm 2.6$ |
| | 1 day (light) | $0.29 \pm 0.05$ | $-0.29 \pm 0.20$ | $-2.92 \pm 0.35$ | 0.922 | 0.259 | NR |

The values of k indicated that in the absence of a dark period after the UV irradiation, the UV resistance by *E. coli* ATCC 8739 and native *E. coli* was similar, with values of 0.54 and 0.52 cm$^2$ mJ$^{-1}$, respectively. Similarly, the maximum inactivation level achievable by the UV system, determined by the parameter $S_{res}$, was similar for both organisms, obtaining more than 4 log-reductions, an inactivation greater than 99.99%. In practice, inactivation kinetics parameters such as *k* do not provide a direct clue about the treatment efficacy; in this context, the dose required to achieve "n" log-reductions is a parameter that allows the direct comparison between different experimental configurations, even if they follow other inactivation kinetics models. In this study, 3 log reductions were achieved in most of the cases. Thus, the parameter $D_3$ was calculated as an indicator of the treatment inactivating efficacy, obtaining a similar value for both ATCC 8739 and native *E. coli* (Table 2).

### 3.2. Effect of the Post-Treatment Incubation on the Treatment Efficacy

The inactivation curves (Figure 5) indicate that the data series obtained after a 24 h incubation in an illuminated environment shows lower inactivation concerning the samples measured directly upon the irradiation and the samples incubated for 24 h in a dark environment. This fact indicates the existence of photoreactivation in the irradiated organisms.

The inactivation kinetics parameters (Table 2) allow for quantitatively addressing the importance of photoreactivation in both studied organisms. Log (S0) values are in the range of $0.08 \pm 0.09$ and $-0.29 \pm 0.20$, indicating that the possible mortality throughout incubation was negligible compared to the inactivation caused by the UV treatment. In the case of *E. coli* ATCC 8739, the post-treatment incubation did not cause a change in *k*. In contrast, the Log ($S_{res}$) was slightly greater after the 24 h incubation in an illuminated environment. On the other hand, in the case of the native *E. coli*, the illuminated incubation reduced the *k* from 0.52 to 0.29 cm$^2$ mJ$^{-1}$. It reduced the maximum level of inactivation reachable from 4.45 to 2.92 log-reductions. In this sense, the values of $D_3$ obtained for *E. coli* ATCC 8739 were similar for the three post-treatment conditions tested, whereas the incubation in an illuminated environment caused 3 log-reductions that could not be achieved in the treatment of native *E. coli* due to tailing. Therefore, the storage of the irradiated organisms under illuminated conditions affects the inactivation rate. Additionally, it influences the maximum level of inactivation achievable by the UV device.

### 3.3. Percentage of Photoreactivation of the Different Bacterial Strains

The (PRP) was determined for both organisms and correlated with UV dose according to first-order kinetics (PRP = a · e$^{\text{b·UV dose}}$). PPR decreased with the UV dose applied (Figure 6) in both cases. Strain ATCC 8739 ranged from 0.01% to 2%. On the other hand, PRP is from 0.2 to 6% for native *E. coli*.

According to the regression parameters determined, for the case of *E. coli* ATCC 8739, once the 3 log-reductions are reached with a dose of 12.8 $\pm$ 1.3 mJ cm$^{-2}$, there is a risk that 0.19% of the initial bacterial concentration will be reactivated at 24 h after the inactivation treatment if the irradiated organisms were stored in an illuminated environment. In the case of native *E. coli*, it is more critical since once achieved 3 log-reductions with a UV dose

of $12.6 \pm 2.0$ mJ cm$^{-2}$, the 1.54% of the initial bacterial concentration at 24 h would be reactivated. Although these percentages may seem small, they represent the reactivation of a noticeable number of organisms when dealing with high bacteria concentrations. The correlation between the PRP and the UV dose applied indicates that using high UV doses not only reduces the bacterial concentration but also prevents the reactivation of a greater fraction of irradiated organisms.

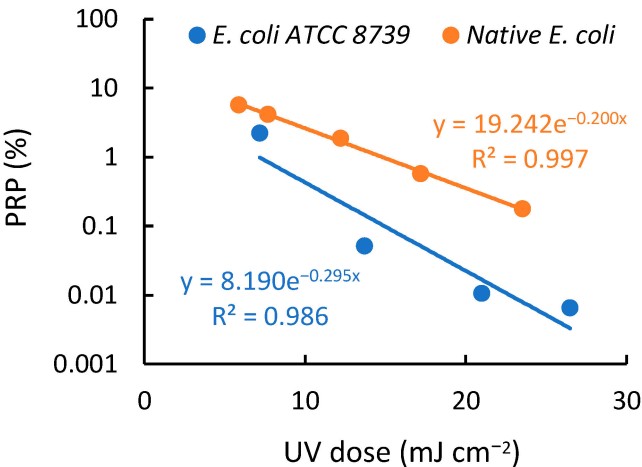

**Figure 6.** Relationship between the percentage of photoreactivation and UV dose for *E. coli* bacterial strains tested.

## 4. Discussion

### 4.1. Native Bacteria vs. ATCC Isolates

In this study, native bacteria naturally occurring in an environment are referred to as native bacteria. It was denoted as E. coli native to maintain the same denomination compared with other studies [27,30,31]. Native bacteria represent the natural diversity of microorganisms present in a particular ecosystem. Moreover, being in their original habitat, they adapt to the conditions in that specific ecosystem. On the other hand, ATCC isolates may represent a subset of that diversity or specific strains chosen for their unique characteristics or research interest. The results obtained in this study allow a comparison of the efficacy of UV disinfection between the two strains and several studies that have worked with the certified strain.

### 4.2. Kinetic Model for E. coli

The design and optimization of UV reactors can be affected by the existence of tails and shoulders in the kinetic curves; the causes of both phenomena have been a matter of debate [15]. Kinetic models with tails describe the inactivation of microorganisms that do not follow first-order kinetics; i.e., their inactivation rate is not constant and decreases with time. In the case of *E. coli*, some studies have found that its inactivation rate is not constant and decreases after a certain time of exposure to UV radiation [19]. This may be due to several factors, such as the formation of cell aggregates that protect the bacteria from UV radiation, the presence of cells at different stages of the growth cycle, and the heterogeneity of the bacterial population. Therefore, tailed kinetic models can be useful to describe the inactivation of *E. coli* by UV radiation and predict its inactivation rate under different experimental conditions. In our case for both strains, in all three treatments (inactivation and reactivation in the light and the dark), they present tailing phenomena at high UV fluence, within the observed UV range (Figure 5); on the other hand, no shoulders were observed in the two strains.

Furthermore, it is well known that reactor configuration influences inactivation kinetics. In previous studies using *E. coli* ATCC 11229 in buffered distilled water and irradiation with conventional low-pressure mercury lamps, there was a dependence between the

occurrence of shoulder and tail with respect to the type of UV device used. Inactivation curves obtained using a collimated beam reactor lacked a shoulder. Still, they showed tails, whereas curves obtained with a continuous flow reactor showed a shoulder, and no tails were observed in the experimental UV dose range up to 30 mJ cm$^{-2}$ [32]. In addition, reactor configuration can influence kinetic models with tails or shoulders for *E. coli*, as the presence of dead zones, hydraulic short circuits, variations in UV intensity, and other factors can affect disinfection efficiency [33]. In our case, it was not to characterize the presence of shoulders at low dose ranges because the reactor did not allow manipulating shorter biases for higher experimental flow rates. On the other hand, the presence of tailing in the kinetics is probably attributed to more resistant cells circulating through the system and not receiving an adequate UV dose, as well as to the experimental reactor configuration itself.

### 4.3. Kinetic Modeling Parameters

With the review of previous studies and the obtained results, it is evident that the UV doses for the inactivation of *E. coli* ATCC 8739 with a continuous flow reactor can vary depending on several factors, such as the type of reactor used, the intensity of UV radiation, and the initial concentration of *E. coli* in the water sample, among others [15,34,35]. In addition, native *E. coli* strains can present different characteristics and resistances that can affect the effectiveness of UV radiation; the present study presents the dose necessary for the inactivation of a native strain of *E. coli* and compares it against a laboratory strain. Comparing with other studies that looked at the sensitivity of *E. coli* bacteria, they found doses between 20 and 40 mJ cm$^{-2}$ for a $-1$ log CFU mL$^{-1}$ and between 50 and 110 mJ cm$^{-2}$ for a $-4$ log CFU mL$^{-1}$, as dose ranges reported for collimated batch reactors [33]. Other studies with collimated reactors report a dose of 6.5 and 19 mJ cm$^{-2}$ for inactivation of $-4$ log CFU mL$^{-1}$ for *E. coli* ATCC 8739 and native-type strains (Harris et al., 1987 [36]). In our case, we obtained inactivation from 7.3 to 14 mJ cm$^{-2}$ for a $-2$ log CFU mL$^{-1}$ and between 7.8 to 12.8 mJ cm$^{-2}$ for a $-3$ log CFU mL$^{-1}$ in the three treatments, inactivation, and subsequent reactivation (light and dark) in the two cases (Table 2).

Because of the diversity of results found, it is usually useful to use a parameter that combines the parameters of the kinetic model ($S_{res}$ and $k$), which for *E. coli* usually turns out to be the $D_4$ (lethal dose for 99.99% of microorganisms) normally used for this purpose. The reviewed studies provide different $D_4$ values for the inactivation of *E. coli* by UV radiation; one study reported a $D_4$ of 16 mJ cm$^{-2}$ for *E. coli* in water treated with UV radiation at a wavelength of 254 nm [37]. Another study reported a $D_4$ of 15 mJ cm$^{-2}$ for *E. coli* in water treated with UV radiation at a wavelength of 254 nm [38]. A third study reported a $D_4$ of 4.4 mJ cm$^{-2}$ for *E. coli* in water treated with UV radiation at a wavelength of 254 nm [39]. In the present study, $D_4$ reduction was not achieved in all cases; this may be due to the reactor configuration, which affects the exposure time required to achieve the same UV dose and the distribution of UV-C light intensity along the exposed volume of water.

### 4.4. Reactivation Processes for E. coli

It is important to study the photoreactivation of *E. coli* because it may have practical implications for water disinfection. While UV radiation can be effective for inactivation, photoreactivation may decrease treatment efficacy if bacteria exposed to UV radiation have the opportunity to recover and repair damage to their DNA. Therefore, it is important to understand the magnitude and duration of photoreactivation to determine the dose of UV radiation needed to ensure adequate water disinfection. The results demonstrated the magnitude of photoreactivation; the percentage of photo-reactivation was dependent on the UV dose applied, which ranged from 4.5 to 26.5 mJ cm$^{-2}$, with the highest percentage of photoreactivation being achieved by native *E. coli* with a value of 5.7% for an applied dose of 5.9 mJ cm$^{-2}$, values similar to those reported in another study [32]. In this study, the irradiated samples were exposed to light in a culture chamber to maximize the chances of photorepair; it is shown that the photoreactivation rate depends on the storage environment of the irradiated water [18,40].

On the other hand, the study of dark reactivation of *E. coli* is important because it allows us to understand better the ability of the bacterium to repair the damage caused by UV radiation in the absence of light. This is relevant because, in certain situations, such as in groundwater or wastewater treatment systems, sunlight cannot penetrate. Therefore, bacteria may be exposed to UV radiation in dark conditions. In both cases, dark repair was absent within 24 h after UV irradiation, the same as in a previous study [20].

Considering that native *E. coli* exhibits higher inactivation resistance and higher photoreactivation capacity versus *E. coli* strain ATCC 8739, it is important to consider these differences when designing water and food disinfection and treatment strategies. It is possible that the doses of UV radiation required to completely inactivate native *E. coli* strains are higher than those required for strain ATCC 8739, which may require adjustments to treatment and disinfection systems. In addition, the photoreactivation ability of native strains may also have implications for the efficacy of water treatment systems using UV radiation, as the recovery of viable bacterial cells may occur in the presence of visible light. Therefore, understanding the differences in resistance and photoreactivation of different *E. coli* strains is essential to ensure the efficacy of disinfection treatments and protect public health.

## 5. Conclusions

The data obtained in this study revealed that the inactivation caused by the irradiation with a flow-through reactor with emission at 254 nm was similar for both the pure *E. coli* ATCC 8739 strain and the native *E. coli*. In this regard, 3 log-reductions were obtained with the application of 13 mJ cm$^{-2}$ for both organisms. However, the post-treatment conditions have a certain influence on the treatment outcome. In the case of storing the irradiated organisms under a dark environment, reactivation was not observed in the 24 h following the treatment. In the case of storing the irradiated organisms under an illuminated climate, the photoreactivation caused a loss of the treatment efficacy. A correlation was observed between the percentage of photoreactivated organisms and the UV dose applied. This implies that the application of a higher UV dose, besides achieving a greater inactivation level, prevents the photoreactivation of a larger fraction of organisms. After 3 log-reductions of the initial bacterial concentration, the photoreactivation caused the reactivation of 0.19% of the inactivated *E. coli* ATCC 8739 and 1.54% of the inactivated native *E. coli*. Although these are small ratios of reactivated organisms, they can determine the success or the failure of a UV treatment if the treated water is stored under a light environment.

**Author Contributions:** Conceptualization: P.D.-S., V.P.-V. and L.R.-M., methodology: P.D.-S., V.P.-V. and L.R.-M., investigation: P.D.-S., resources: P.D.-S., writing—original draft preparation: P.D.-S.; writing—review and editing: V.P.-V., L.R.-M., E.S.-C. and E.S. All authors have read and agreed to the published version of the manuscript.

**Funding:** This research has been funded by the Salesian Polytechnic University research funds, project number 010-005-2021-07-01.

**Institutional Review Board Statement:** Not applicable.

**Informed Consent Statement:** Not applicable.

**Data Availability Statement:** Not available.

**Conflicts of Interest:** The authors declare that they have no known competing financial interest or personal relationships that could have appeared to influence the work reported in this paper.

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
