# Peer review of "Comparative Study of UV Radiation Resistance and Reactivation Characteristics of E. coli ATCC 8739 and Native Strains: Implications for Water Disinfection"

_sustainability, doi:10.3390/su15129559_

Round 1
Reviewer 1 Report
The article by Duque-Sarango et al. investigated the comparative effects of UV radiation on two strains of Ecoli, a technology that is widely used to treat wastewater. Although there are several studies on the effect of UV on bacteria, there are fewer focusing on the reactivation of bacteria after UV, especially in the dark. Therefore, the findings are considered interesting. The paper is generally written well in the results and discussion section, but the introduction has a few instances where the authors need to check/verify the data and revisit the sentence formation with an evaluation of grammar. The methods section contains enough details to repeat the experiments and the discussion is well structured with a comparison with previously published articles. The discussion also mentions the novelty of the study and how this could potentially advance the field.
Below are a few of the points to consider for revision:
Line 42: please specify what type of water resources…’groundwater/freshwater’?
Line46: please specify what type of water needs to be treated, since not all water requires it.
Lines 50-54 needs reference citation
Line 57: reuse of wastewater effluent is not an alternative to water demand. Please correct the sentence formation to correctly portray the idea.
Line 58: countries do not use ‘wastewater’ for irrigation, they probably use it after treatment. Please verify.
Line 59: Specify what “practice’
Line 65: 200 CFU/ 100ml. Please check
Line 68-69: not all Ecoli are infectious.
Line 86: remove [period’ after reference 14
Line89: please check sentence formation
Line 136: membrane filtration method needs a reference
Line 157: please check for a updated version of the guidelines, the one referenced here was published back in 1986.
Figure 4: Please add details in the legend. What does the blue and black containers denote? Additionally add these details in the text (line 190 and 191).
Line 240: please check grammar/sentence formation
Line 311: please check grammar/sentence formation
Line 354, 363: ‘study’ instead of ‘studies’
The paper is generally written well in the results and discussion section, but the introduction has a few instances where the authors need to check/verify the data and revisit the sentence formation with an evaluation of grammar.
Reviewer 2 Report
I have some suggestions to improve their work.
A grammatical revision is required throughout the text; for example, some articles, commas, adverbs, Etc., must be included. The authors have to make some modifications.
Throughout the text, check the conjugation of the verbs.
Check some words, for example, thermo-resistant, etc.
Check the order of ideas in some phrases; for example, line 80 – 83 is better if you write: In most cases, disinfection is the last stage in drinking water treatment, and its purpose is to eliminate pathogenic microorganisms that cause waterborne diseases.
The paper needs to be checked carefully for typos and grammatical errors.
The scientific names should be written in italics. Correctly spell E. coli.
Keep the same font style and size throughout the manuscript.
In the reference list, some species names are not italics (even in the text, check this aspect). Also, some references are duplicated, for example, 20 and 21.
The bacteria name E. coli and the other microorganisms in references it is misspelled.
The abstract count is 267 words and, according to Instructions for Authors, should be shortened to 200 words maximum.
In the discussion section, you can answer the next questions
Why did you decide to name bacteria isolated from wastewater as native bacteria?
What are the differences between the native bacteria and the ATCC isolate?
The research is very interesting and well-written, but minor editing of the English language is required.
A grammatical revision is required throughout the text; for example, some articles, commas, adverbs, Etc., must be included.
Reviewer 3 Report
Overall, the article is well written linguistically and easy to read. The experimental procedures are technically sound, extensively described and discussed, with additional figures to explain the methodological procedure. The results were also presented in clear figures, appropriately discussed and compared to current literature.
